# Genomics of *Staphylococcus aureus* Strains Isolated from Infectious and Non-Infectious Ocular Conditions

**DOI:** 10.3390/antibiotics11081011

**Published:** 2022-07-27

**Authors:** Madeeha Afzal, Ajay Kumar Vijay, Fiona Stapleton, Mark D. P. Willcox

**Affiliations:** School of Optometry and Vision Science, University of New South Wales, Sydney, NSW 2052, Australia; v.ajaykumar@unsw.edu.au

**Keywords:** *Staphylococcus aureus*, ocular infectious isolates, whole genome sequencing, virulence factors

## Abstract

*Staphylococcus aureus* is a major cause of ocular infectious (corneal infection or microbial keratitis (MK) and conjunctivitis) and non-infectious corneal infiltrative events (niCIE). Despite the significant morbidity associated with these conditions, there is very little data about specific virulence factors associated with the pathogenicity of ocular isolates. A set of 25 *S. aureus* infectious and niCIEs strains isolated from USA and Australia were selected for whole genome sequencing. Sequence types and clonal complexes of *S. aureus* strains were identified by using multi-locus sequence type (MLST). The presence or absence of 128 virulence genes was determined by using the virulence finder database (VFDB). Differences between infectious (MK + conjunctivitis) and niCIE isolates from USA and Australia for possession of virulence genes were assessed using the chi-square test. The most common sequence types found among ocular isolates were ST5, ST8 while the clonal complexes were CC30 and CC1. Virulence genes involved in adhesion (*ebh*, *clfA*, *clfB*, *cna*, *sdrD*, *sdrE)*, immune evasion (*chp*, *esaD*, *esaE*, *esxB*, *esxC*, *esxD*), and serine protease enzymes (*splA*, *splD*, *splE*, *splF)* were more commonly observed in infectious strains (MK + conjunctivitis) than niCIE strains (*p =* 0.004). Toxin genes were present in half of infectious (49%, 25/51) and niCIE (51%, 26/51) strains. USA infectious isolates were significantly more likely to possess *splC*, *yent1*, *set9*, *set11*, *set36*, *set38*, *set40*, *lukF-PV*, and *lukS-PV* (*p* < 0.05) than Australian infectious isolates. MK USA strains were more likely to possesses *yent1*, *set9*, *set11* than USA conjunctivitis strains (*p =* 0.04). Conversely USA conjunctivitis strains were more likely to possess *set36 set38*, *set40*, *lukF-PV*, *lukS-PV* (*p* = 0.03) than MK USA strains. The ocular strain set was then compared to 10 fully sequenced non-ocular *S. aureus* strains to identify differences between ocular and non-ocular isolates. Ocular isolates were significantly more likely to possess *cna* (*p* = 0.03), *icaR* (*p =* 0.01), *sea* (*p* = 0.001), *set16* (*p* = 0.01), and *set19* (*p =* 0.03). In contrast non-ocular isolates were more likely to possess *icaD* (*p* = 0.007), *lukF-PV*, *lukS-PV* (*p =* 0.01), *selq* (*p* = 0.01), *set30* (*p* = 0.01), *set32* (*p =* 0.02), and *set36* (*p* = 0.02). The clones ST5, ST8, CC30, and CC1 among ocular isolates generally reflect circulating non-ocular pathogenic *S. aureus* strains. The higher rates of genes in infectious and ocular isolates suggest a potential role of these virulence factors in ocular diseases.

## 1. Introduction

*Staphylococcus aureus* is responsible for nearly 70% of ocular infections worldwide [1]. These can result in tissue damage, morbidity, and vision loss [2,3]. *S. aureus* infections involving the cornea (microbial keratitis; MK) can be sight-threatening and the organism is the most common cause of MK in Australia [4,5] and USA [6,7]. *S. aureus* can also cause conjunctivitis [8] and non-infectious corneal infiltrative events (niCIE) during contact lens wear [9]. 

*S. aureus* is known to encode a diverse arsenal of virulence determinants that enables it to cause a variety of infections [10]. The genomic make-up of *S. aureus* influences the virulence of its strains and pathogenicity associated with its disease [11]. The antibiotic susceptibility data of the isolates previously reported [12] and used in this study demonstrated that although most of the strains were multi-drug resistant (MDR), the non-infectious (niCIE) strains were more susceptible to antibiotics (ciprofloxacin, ceftazidime, oxacillin) than were the conjunctivitis strains, and the conjunctivitis strains were more susceptible to antibiotics (chloramphenicol, azithromycin) than were the MK strains [12]. MK strains from Australia were more susceptible to antibiotics (ciprofloxacin, oxacillin) compared to MK strains from USA [12]. Whilst several studies have examined which virulence factors might be involved in the development of keratitis by *S. aureus*, there is much less information on the association of virulence factors with conjunctivitis or niCIE [13]. Similarly, as outlined previously [13], infectious isolates (MK + conjunctivitis) had a higher frequency of genes involved in evasion of the immune system and invasion of the host (*hlg*, *hld*) compared to niCIE strains. On the other hand, *scpA*, that encodes a staphylococcal cysteine proteinase, was more common in niCIE strains. However, those previous studies only examined a subset of genes, specifically those that had been previously reported to be involved in infections of the eye or antibiotic resistance. This current study examines the whole genome of a subset of strains isolated from MK, conjunctivitis, and niCIE. This analysis may identify new genes that are associated with particular infections or resistance to antibiotics.

Whole genome sequencing (WGS) is a widely used technique that can identify antibiotic resistance genes, virulence determinants, emerging bacterial lineages, and their population structures [14,15,16]. Comparative genomics and genome-wide association studies of clinical isolates can reveal genetic determinants that may be important in the setting of specific infections. For example, WGS has been used successfully to examine *S*. *aureus* isolates collected from systemic infections (bloodstream, airways, endocarditis, and joint infections) to further understand specific population structures as well as to explore the relationship between virulence factors and patient outcomes [16,17,18,19]. 

WGS of *S. aureus* strains isolated from different infections (airways, soft-tissues, and skin lesions) showed high level of diversity and co-presence of local, global, livestock-associated, and hypervirulent clones and found that some virulence factors and clones were disease specific. For example, the sequence type ST22 was associated with toxic shock syndrome toxin TSST-1 and ST5 was associated with enterotoxins (SE) [18]. Another study explored genomic relatedness between commensal nasal isolates and those isolated from prosthetic joint infections and found the commensals shared the same clonal complex (CC) and the prevalence of virulence genes among isolates from commensal and prosthetic joint infections in arthroplasty patients was almost equal, suggesting that commensal *S. aureus* nasal clones can cause joint infections [19]. 

In the current study, WGS was used to analyze 25 *S. aureus* strains from ocular infectious and niCIEs isolated from USA and Australia. A custom analytical pipeline determined MLST, to define circulating *S. aureus* ocular lineages in infectious and non-infectious strains from USA and Australia, as well as the presence or absence of 128 known *S. aureus* virulence factors. The ocular strains were then compared to 10 fully sequenced non-ocular strains to determine the key virulence factors involved in ocular diseases.

## 2. Results 

### 2.1. General Features of the Genomes

After de novo assembly, the isolates had different numbers of contigs ranging from 328 for SA31 to 3916 for SA86. Isolates had an average guanine plus cytosine (GC) content of 32.8%. The tRNA copy number for the isolates ranged from 60 to 89. Similarly, the number of coding sequences (CDS), which was determined based on Prokka annotation pipeline, ranged from 2614 (in M19-01) to 3873 (SA86). The general features of isolates are provided in Table 1.

### 2.2. Acquired Antimicrobial Resistance Genes

Eighteen different types of acquired antimicrobial resistance genes for various classes of antibiotics were detected in this study (Table 2). Antimicrobial resistance genes for vancomycin (*vanA*), fusidic acid (*fusA*, *fusB*), trimethoprim (*dfrA*, *dfrB*, *dfrG*), ciprofloxacin (*gyrA*, *gyrlA*, *grlB*), fosfomycin (*fosB*), and rifampin (*rpoB*) were not found in any of the strains. The beta lactamase resistance gene *blaZ* which encodes penicillin resistance was found in 76% of isolates. However, the methicillin resistance gene *mecA* was found in 28% of strains, all of which were from the USA; the possession of *mecA* was significantly more common in infectious isolates from USA than from Australia (*p =* 0.0016). 

The aminoglycoside resistance genes were significantly more common (*p =* 0.0006) in strains from the USA, with only strain M28-01 isolated from MK in Australia possessing one of these genes, *ant(9)-la*. Genes associated with resistance to macrolides, lincosamide, or streptogramin B were significantly more likely to be found in USA isolates (*p* = 0.002), with only Australian isolates M28-01 possessing *erm(A)* and SA25 possessing *erm(A)*, *msr(A)*, and *erm(C)*. Six isolates possessed *tetK* that encodes tetracycline resistance, and these were scattered across isolates from MK (2 USA, 1 Australian), conjunctivitis (1 USA) and niCIE (2 Australia). Resistance gene for tetracycline (*tetM*) and quaternary ammonium compound (*qacD*) were found in single isolate (USA) whereas pseudomonic acid (mupirocin) was present in only two USA isolates and quaternary ammonium compound *qacB* was found in single USA and single Australian isolate. Chloramphenicol resistance gene *cat(pC233)* was only found in a single Australian isolate (Table 2).

Overall, in Australian infectious isolates only five acquired antimicrobial resistance genes were detected. As the current study relied on draft genomes it may not be able to predict actual genomic diversity and could not detect actual antimicrobial resistance genes. There could be more genes, complete gene sequence of isolates can show the actual number of antimicrobial resistance genes. Similarly, USA infectious isolates had acquired 17 different antimicrobial resistance genes (Table 2). NiCIE isolates from Australia had acquired six different antimicrobial resistance genes. One USA infectious isolate, SA101, had the largest number of acquired antimicrobial resistance genes (eight). 

### 2.3. S. aureus Virulence Determinants

Of the 128 virulence factors examined, 22 virulence genes (*atl*, *ebh*, *clfA*, *clfB*, *cna*, *ebp*, *eap*, *efb*, *fnbA*, *fnbB*, *icaA*, *icaB*, *icaC*, *icaD*, *icaR*, *sdrC*, *sdrD*, *sdrE*, *sdrF*, *sdrG*, *sdrH*, *spa*) in VFDB are categorized as genes involved in *S. aureus* adhesion. Of these adhesins, *atl*, *ebp*, *eap*, *efb*, *fnbA*, *fnbB*, *icaA*, *icaB*, *icaC*, *icaR*, *sdrC*, and *spa* were found in ≥96% of all *S. aureus* isolates. On the other hand, *sdrF*, *sdrG*, *sdrH* were not detected in any of the strains.

*S. aureus* strains from ocular infectious and niCIE showed non-significant differences in the frequency of possession of six adhesins (Figure 1), with only possession of *icaD* showing a trend towards being more common in niCIE isolates (*p* = 0.1).

When differences were examined for the possession of adhesins in the infectious isolates from different countries, there were no significant differences observed in MK and conjunctivitis isolates from USA and Australia.

Of the remaining 106 virulence genes, 15 were categorized as enzymes in VFDB. These were genes for the cysteine proteases *scpA* and *sspB*, the serine proteases *sspA*, *splA*, *splB*, *splC*, *slpD*, *splE*, *splF*, hyaluronate lyase *hysA*, the lipases *geh* and *lip*, staphylocoagulase *coa*, staphylokinase *sak* and thermonuclease *nuc*. Of these genes, 67% (9/15; *sspB*, *hysA*, *geh*, *lip*, *v8*, *sspA*, *sak*, and *nuc)* were found in ≥96% of all *S. aureus* isolates. The isolates from infections or niCIEs isolates did not show significant differences (*p* > 0.05) or trend towards significance (*p* = 0.1), for the possession of other seven proteases, (Figure 2). 

Similarly, the frequency of protease genes in strains isolated from MK or conjunctivitis was not significantly different, although conjunctivitis strains (100%) had a trend for more frequent presence of *splA* (*p =* 0.1) and *splF* (*p =* 0.1) than MK strains (58%). In infectious isolates (MK + conjunctivitis) from the USA, possession of *splC* (100% vs. 55%; *p =* 0.03) and *splB* (90% vs. 44%; *p =* 0.05) was higher and there was also a trend for higher possession of *splD* (100% vs. 66%; *p =* 0.08) and *splA* (100% vs. 55%; *p =* 0.1) compared to Australian isolates, except *scpA* (100% vs. 60%; *p =* 0.08) which was higher in infectious isolates (MK + conjunctivitis) from Australia. 

Of the remaining 91 virulence genes, five were involved in immune evasion (IE; *adsA*, *chp*, *cpsA*, *scn*, *sbi*), and 12 genes were involved in the type VII secretion systems (*esaA*, *esaB*, *esaD*, *esaE*, *esaG*, *essA*, *essB*, *essC*, *esxA*, *esxB*, *esxC*, *esxD*). Of these, 10/17 (*adsA*, *cpsA*, *scn*, *sbi*, *esaA*, *esaG*, *essA*, *essB*, *essC*, and *esxA*) were found in ≥96% of all *S. aureus* isolates. There were no significant differences or trends in possession of any IE or type VII secretion system genes by disease group or by country. Figure 3 shows the differences in possession of seven of these genes between infectious and niCIE isolates.

The remaining 74 virulence genes encoded for toxins including hemolysins (_hla, hlb, hld, hlgA, hlgB, hlgC_), enterotoxins (*sea*, *seb*, *sec*, *sed*, *see*, *seg*, *seh*, *sei*, *sej*, *yent1*, *yent2*, *selk*, *sell*, *selm*, *seln*, *selo*, *selp*, *selq*, *selr*, *selu*), exfoliative toxins (*eta*, *etb*, *etc*, *etd*), exotoxins, also known as enterotoxin like genes, (*set1*, *set2*, *set3*, *set4*, *set5*, *set6*, *set7*, *set8*, *set9*, *set10*, *set11*, *set12*, *set13*, *set14*, *set15*, *set16*, *set17*, *set18*, *set19*, *set20*, *set21*, *set22*, *set23*, *set24*, *set25*, *set26*, *set30*, *set31*, *set32*, *set33*, *set34*, *set35*, *set36*, *set37*, *set38*, *set39*, *set40*), leukocidins (*lukF-like*, *lukM*, *lukD*, *lukE*, *lukf-PV*, *lukS-PV*), and toxic shock syndrome toxin (*tsst*). 

Of these, the hemolysins *hla*, *hlgA*, *hlgB*, *hlgC* were found in ≥96% of all *S. aureus* isolates. Of the remaining 70 toxins, *sed*, *see*, *sej*, *selp*, *selr*, *eta*, *etb*, *etc*, *etd*, *set10*, *set12*, *set14*, *set20*, *lukM* were not detected in any of the isolates, and *hlb*, *sell*, *set35*, *lukF-like*, *lukE* were present only in 4% of all *S. aureus* isolates. However, 51 toxins showed some differences between *S. aureus* infectious and niCIE isolates (Figure 4). Of these the only significant differences or trends for differences were as follows: niCIE isolates tended to have a higher frequency (50%) of only *set3* (*p =* 0.1) (Figure 4) than infectious isolates (16%), and infectious isolates tended to have a higher frequency (95%) of only *hld* (*p =* 0.1) than niCIE (67%) (Figure 5). 

Overall conjunctivitis strains were more likely to possess *set36* (43% vs. 0%; *p* = 0.03), *set38* (57% vs. 8%; *p =* 0.03), *set40* (43% vs. 0%; *p =* 0.03), *lukF-PV* (43% vs. 0%; *p =* 0.03), *lukS-PV* (43% vs. 0%; *p =* 0.03), with a trend for *set31* (57% vs. 17%; *p =* 0.1) than MK strains. These 51 toxins were also examined for differences in the isolate’s country of origin. The only differences were for MK strains, where isolates from the USA had a significantly higher frequency of possession of *yent1* (60% vs. 0%; *p =* 0.04), *set9* (60% vs. 0%; *p* = 0.04), and *set11* (60% vs. 0%; *p* = 0.04) than MK isolates from Australia.

The VFDB results of these 25 ocular isolates were compared with previously published non-ocular isolates for the possession of the 128 virulence determinants. Eight genes involved in adhesion (*ebp*, *eap*, *efb*, *fnbA*, *fnbB*, *icaA*, *icaR*, *sdrC*) were found in all ocular and non-ocular *S. aureus* isolates and three (*sdrF*, *sdrG*, *sdrH*) were not found in any isolate. *S. aureus* ocular isolates showed higher frequency for the possession of *cna* (40% vs. 0%; *p =* 0.03) and *icaR* (100% vs. 70%; *p =* 0.01) whereas non-ocular isolates showed higher frequency for the possession of *icaD* (60% vs. 12%; *p =* 0.007), *ebh* (90% vs. 60%; *p =* 0.1), and *sdrD* (100% vs. 68%; *p =* 0.07). Of 15 enzymes, 9 (*spa*, *sspB*, *sspC*, *hysA*, *geh*, *lip*, *sspA*, *coa*, *nuc*) were found in all *S. aureus* isolates and no significant differences (or trends) were found in the possession of any other enzyme-associated gene. Similarly, all five (*adsA*, *cpsA*, *scn*, *sbi*, *chp*) genes involved in immune evasion were found in all isolates. Six (*esaA*, *esaB*, *esaG*, *essA*, *essB*, *esxA*) of the genes involved in type VII secretion system were found in ≥96% of all *S. aureus* isolates, with non-significant differences in frequency of possession of the remaining six, type VII secretion system gene, *esaD* (80% vs. 100%; *p =* 0.29), *esaE* (80% vs. 100%; *p =* 0.29), *essC* (92% vs. 100%; *p*
*=* 0.99), *esxB* (76% vs. 100%; *p* = 0.15), *esxC* (76% vs. 100%; *p* = 0.15), *esxD* (76% vs. 100%; *p* = 0.15) in ocular and non-ocular isolates respectively. 

Of 74 toxins, four hemolysin genes (*hla*, *hlgA*, *hlgB*, *hlgC*) were found in ≥96% of all *S. aureus* isolates and 18 toxin genes (*hlb*, *sed*, *see*, *sej*, *selp*, *selr*, *eta*, *etb*, *etc*, *etd*, *set10*, *set12*, *set14*, *set20*, *set35*, *lukF-like*, *lukM*, *lukE*) were found in ≤4% of all strains. Of the remaining 52 toxins, *S. aureus* ocular isolates possessed sea (80% vs. 20%; *p* = 0.001), *set1* (28% vs. 0%; *p* = 0.08), *set5* (32% vs. 0%; *p* = 0.07), *set16* (44% vs. 0%; *p* = 0.01), *set19* (36% vs. 0%; *p* = 0.03), whereas non-ocular isolates possessed *selq* (30% vs. 0%; *p* = 0.018), *set30* (70% vs. 24%; *p* = 0.01), *set32* (50% vs. 12%; *p* = 0.02), *set36* (50% vs. 12%; *p* = 0.02), *set37* (100% vs. 72%; *p* < 0.0001), *lukD* (100% vs. 72%; *p* < 0.0001), *lukF-PV* (60% vs. 16%; *p* = 0.001), and *lukS-PV* (60% vs. 16%; *p* = 0.001). 

### 2.4. Sequence Types and Clonal Complexes of S. aureus Isolates

The MLST typing of 25 *S. aureus* genomes revealed a total of 14 distinct sequence types (STs) and seven clonal complexes (Table 3), ST5 (n = 5, 20%) and ST8 (n = 4, 16%) were the most common sequence types in this cohort of ocular isolates. For strain M19-01, no ST type was identified, and was named as NI (Table 3). In the current study most of the USA isolates were from CC5 or CC8, whereas there was a greater spread of sequence types and clonal complexes in the Australian isolates. 

The core, shell (genes present in two or more strains), and pan genes of published isolates are provided in Appendix A. The core genes were used to create a phylogenetic tree of the *S. aureus* isolates using *S. aureus* NCTC 8325 (NC_007795.1) as a reference strain. The ten published non-ocular *S. aureus* isolates downloaded from the Genebank database were also included. Isolates of the same clonal complex or same sequence type were grouped together in the same cluster irrespective of their ocular condition or country of origin (Figure 6). The core genomes formed three groups in the phylogenetic tree (Figure 6). Isolates in Group 1 were related, as they belonged to the same CC5. This group also contained all the extensively-drug resistant isolates (XDR: resistant to almost all antibiotics) (SA111, SA112, SA113) and three multi-drug resistant (MDR: resistant to three different classes of antibiotics) isolates (SA90, SA48, SA46) reported in the previous study [12]. Isolates from CC30 in Group 2 were further clustered into two sub-lineages based on their core genes and STs. Group 3 was larger and contained the majority of MDR strains. Australian strains clustered into sub lineages within group 3, whereas USA isolates with same ST8 clustered together within group 3. 

The pan phylogenetic relationships of these *S. aureus* isolates were assessed (Figure 7). This divided the *S. aureus* isolates into three major groups. Group 1 of the pan genome phylogeny contained only two isolates, ocular isolate M43-01 and a non-ocular isolate of the same clonal complex. The second group included isolates with the same clonal complex, and XDR (resistant to almost all antibiotics) and MDR (resistant to three different classes of antibiotics) strains, irrespective of their ocular condition and country of origin. Group 3 was further divided into two subgroups; isolates with the same pangenome and CC or ST were clustered together. Isolates in group 3 had a large number of pan genes. Isolates belonging to the same sequence type or clonal complex were grouped together, for example, isolates 129, 31, 114, 27 were from clonal complex 30.

## 3. Discussion

This study investigated genomic differences in resistance and virulence genes of *S. aureus* isolates from different infectious (MK and conjunctivitis) and non-infectious (niCIE) ocular conditions from USA and Australia. Based on previous phenotypic susceptibility [12] and PCR data [13] it was expected that there would be differences in the resistance and virulence determinants between infectious and non-infectious disease. Most (n = 22, 88%) of the isolates used in the study were MDR [12]. Phenotypically non-infectious (niCIE) isolates in a previous study were more susceptible to antibiotics (ciprofloxacin, ceftazidime and oxacillin) than conjunctivitis and MK strains, and MK isolates from USA were more resistant to antibiotics than MK isolates from Australia (ciprofloxacin, ceftazidime and oxacillin) [12]. The current study’s genotypic data shows that infectious isolates from USA harbored more antimicrobial resistance genes (ARGs) compared to Australian isolates, which supports the phenotypic data of the previous study. Similarly, PCR results for a subset of 12 known virulence genes had previously reported that genes involved in evasion and invasion (*hlg* and *hld*) were more commonly found in infectious isolates than niCIE [13]. The current study results for *hld* are consistent with the previous report [13]. However, *hlg* which was more common in infectious strains than niCIE strains in the previous study [13] was found in ≥96% of all *S. aureus* isolates in the current study. In addition, due to the selection of isolates in the current study, staphylococcal cysteine proteinase *scpA*, which was more common in niCIE isolates than infectious strains [13] in the previous study, whilst being more commonly observed in niCIE strains than infectious strains in the current study, did not reach significance (100% vs. 79%, *p* = 0.28).

Overall, 76% of all strains possessed the acquired penicillin resistance gene *blaZ* but only 28% of strains, all from USA infectious (MK+ conjunctivitis), possessed *mecA* (i.e., were MRSA). The high level of MRSA among *S. aureus* ocular isolates from USA in the current study is consistent with previous studies [20,21]. The current study reports low level of MRSA among ocular isolates from Australia which supports previous studies showing low rates (≤6.3%) of MRSA among *S. aureus* ocular isolates from Australia [12,22]. 

The aminoglycoside resistance genes *aac (6′)* and *aph (2′)*, which encode for gentamicin resistance, were found in only one isolate from USA which is consistent with phenotypic susceptibility [12] and other previous studies from USA and Australia [22,23,24] which suggest gentamicin remains a good option to treat *S. aureus* ocular infections in both Australia and USA. Genes *ant (6)-la*, *aph (3′) III*, which encode for streptomycin resistance, were found in three USA isolates but in none of the Australian isolates. Streptomycin is no longer used in clinical treatment [25], so this resistance may not be clinically relevant but does suggest environmental selection for the persistence of genes. Gene *ant (9*)-la, which confers resistance to spectinomycin, was found in four USA isolates (three were MRSA) and one isolate from Australia. Several previous reports showed an association between aminoglycoside resistance and methicillin resistance [26,27]. Gene *aadD*, which is responsible for resistance to kanamycin/neomycin and tobramycin [28], was found in four USA isolates. 

Strains from niCIE showed a trend of higher frequency possession of *icaD*, the intercellular adhesion gene, is involved in biofilm production [29]. As niCIE are associated with contact lens wear and contact lenses may provide a surface where bacteria can attach and colonize as a biofilm [30], it is perhaps not surprising that possession of *icaD* was more common in niCIE isolates and suggests that biofilm formation mediated by this gene is not critical for ocular surface infection (i.e., MK or conjunctivitis). In the current study when ocular isolates were compared to non-ocular isolates, they showed higher frequency for the possession of *cna* (40% vs. 0%; *p =* 0.03) and *icaR* (100% vs. 70%; *p =* 0.01), whereas non-ocular isolates showed higher frequency for the possession of *icaD* (60% vs. 12%; *p =* 0.007), *ebh* (90% vs. 60%; *p* = 0.1), and *sdrD* (100% vs. 68%; *p =* 0.07). The product of *cna*, collagen binding adhesin, has been reported to be involved in the pathogenesis of *S. aureus* keratitis [31] and the possession of this gene in ocular strains in the current study confirms that it may be an important virulence determinant in *S. aureus* ocular infections. Gene *icaR* is a strong negative regulator of biofilm formation, and its absence enhances PNAG (poly-N-acetylglucosamine) production and biofilm formation [32,33]. Ocular strains used in this study are enriched with *icaR* which further suggests biofilm formation is not an absolute requirement for *S. aureus* ocular infections. 

Non-ocular strains in the current study were enriched with *icaD* which is involved in biofilm production [29]; this suggests that biofilm formation is important for their non-ocular pathogenesis. Gene *ebh* is a cell wall-associated fibronectin binding protein [34] which helps *S. aureus* to adhere to host extracellular matrix (ECM) and plays a role in cell growth, envelope assembly [35] while contributing to structural homeostasis of bacterium by forming a bridge between the cell wall and cytoplasmic membrane [36]. The lower frequency of *ebh* possession in *S. aureus* ocular strains suggests it has a minor role in eye infections. Gene *ebh* is produced during human blood infection, as serum samples taken from patients with confirmed *S. aureus* infection were found to contain anti-*ebh* antibodies. Gene *sdrD* (serine–aspartate repeat protein D) is member of the MSCRAMMs (microbial surface components recognizing adhesive matrix molecules) [37], promotes the adherence of *S. aureus* to nasal epithelial cells [38], human keratinocytes [39], and contributes to abscess formation [40]. A high prevalence of *S. aureus sdrD* gene is reported among patients with bone infections [41] which suggests that *sdrD* may contribute to systemic infection. *sdrD* is also reported to aid the pathogen in immune evasion by increasing *S. aureus* virulence and survival in blood [40]. The lower frequency of possession of this gene in ocular isolates indicates *sdrD* may not be involved in pathogenesis of currently prevalent types of eye infections, however, *S. aureus* with *sdrD* could contribute more to eye infection.

Overall, isolates from conjunctivitis had a higher frequency for possession of the serine proteases *splA* and *splF* than isolates from MK. Infectious isolates from USA were significantly, or trended to be more likely to, possess the proteases *splC splA*, *splB*, and *splD* than infectious isolates from Australia. Serine proteases are encoded on the νSaβ pathogenicity island [42,43]. The *spl* operon is present in most of *S. aureus* strains but some strains may not have the full operon [44]. Previous studies suggest that serine proteases are expressed during human infections and modulate *S. aureus* physiology and virulence [45], but their role in ocular infections is unknown. The current study suggests that some of these serine proteases may have a role in pathogenesis of conjunctivitis, and this should be studied in future experiments. Again, the trend of pathogenic isolates from USA infections to possess other serine proteases might be related to the different clonal types circulating in the USA. Studies reported ST5 (27%), ST8 (16%), ST30 (9%), and ST45 (6%) as prevalent clonal types in USA ocular isolates [46]. However, a study from tropical northern Australia reported CC75 as a prevalent clone in Australia [47]. In the current study 50% of infectious strains possessing serine proteases were CC5 (ST5, ST105, and ST840), 30% were CC8 (ST8), and 10% were CC15 (ST15) and CC30 (ST30). All strains from CC5 and CC30 possessed 4−6 proteases whereas strains from CC8 and CC15 possessed all six proteases. There was a greater spread of sequence types and clonal complexes in the Australian infectious isolates. 

There were several differences in possession of toxins genes of the *set* family and others. The *set* genes are similar to staphylococcal superantigens but more likely to be involved in immune avoidance [48]. Several toxins in staphylococci are often carried on large mobile genetic elements (MGEs) known as pathogenicity islands that can be horizontally transferred [49] and can be located on the pathogenicity island SaPIn2, SaPIl, and SaPIboy [50]. An increasing number of enterotoxins and enterotoxin-like genes in *S. aureus* have been identified and it is a global trend that around 80% of *S. aureus* both pathogenic and non-pathogenic isolates carry an average of 5−6 enterotoxin genes [51,52,53]. Whether the *set* genes in different strains were present on pathogenicity islands will be examined in future studies.

The current study’s finding that enterotoxin E (*sea)* was more commonly found in ocular strains, infectious strains from USA (70%), AUS (33%), niCIE strains (50%) than non-ocular strains (20%), however, the current study findings are not consistent with earlier studies [54,55]. *S. aureus* strains isolated from atopic patients experiencing keratoconjunctivitis with corneal ulceration, possessed enterotoxins more frequently compared to patients with no ulceration [56]; the role of enterotoxins in ocular infections remains to be fully defined. Another study found enterotoxin and enterotoxin-like genes were found to be highly correlated with MRSA and predictive for MDR status in ocular isolates [57]. The antibiotic susceptibility data of the isolates used in the current study shows that most of the strains were multi-drug resistant (MDR) [12], so the distribution of enterotoxin-like genes in 28% of MRSA USA infectious strains (MK + conjunctivitis) may indicate their MDR status, but the presence of enterotoxin-like genes in MSSA (methicillin sensitive *S. aureus*) MDR strains from other conditions indicates enterotoxin genes are probably associated with the source of isolation. The genes *lukF-PV* and *lukS-PV* encode for the Panton–Valentine leukocidin which is linked to community acquired MRSA infections [58]. Their presence in conjunctivitis strains from USA (60%) in the current study supports previous studies which reported that Panton–Valentine leukocidin (*pvl*) is found in the majority of (67%) ocular strains [59].

The finding that there were no differences in the possession of genes related to immune evasion or type VII secretion systems (*adsA*, *chp*, *cpsA*, *scn*, *sbi*, *esaA*, *esaB*, *esaD*, *esaE*, *esaG*, *essA*, *essB*, *essC*, *esxA*, *esxB*, *esxC*, *esxD*) between different isolate types, countries, or ocular and non-ocular strains, and the finding that most isolates possessed the adhesin genes *atl*, *ebp*, *eap*, *efb*, *fnbA*, *fnbB*, *icaA*, *icaB*, *icaC*, *icaR*, *sdrC*, and *spa*, the enzyme genes (proteases, thermonuclease, lipase, staphylokinase, and hyaluronate lyase) *sspB*, *scpA*, *hysA*, *geh*, *lip*, *v8*, *sspA*, *sak*, and *nuc*, and the hemolysin genes *hla*, *hlgA*, *hlgB*, *hlgC* might indicate that possession of these genes is important either for survival in either the eye or in the environment prior to gaining access to the ocular surface to cause infection or inflammation. 

The core and pan genome phylogenies included strains from all ocular conditions. Acquired genes are part of the pan rather than core genome [60] and the presence of larger pan genomes points towards the acquisition of new genes [61]. The core genome (which is almost 90% of pan or total genome) refers to the conserved genes present in a species, which might differ in each individual strain within that species [62]. With respect to multi-locus sequence typing, in the current study sequence types (STs), ST5 (20%), ST8 (16%), clonal complex (CC), CC30 (16%), and CC1 (12%) were the most prevalent (predominant) types respectively. Previous studies identified that specific lineages including ST5 and ST8 are common among *S. aureus* ocular strains [54,59,63,64,65]. ST5 MRSA isolates from USA were the frequent cause of hospital acquired infections [66] and ST8 MRSA isolates from USA most commonly the causes of community acquired skin and soft tissue infections [67,68]. This suggests that *S. aureus* isolates from ocular infections align with major circulating pathogenic *S. aureus* strains capable of causing systemic infections.

## 4. Material and Methods

### 4.1. Bacterial Isolates

Twenty-five *S. aureus* ocular isolates, 9 isolated from infections (MK + conjunctivitis) in Australia, 10 isolated from infections (MK + conjunctivitis) in USA and 6 isolated from niCIEs in Australia were used (Table 4). The isolates were selected from a larger collection of strains based on their published susceptibility to various antibiotics [12] and possession of virulence genes [13]. Most strains were multi-drug resistant (MDR; Table 4). 

### 4.2. Whole Genome Sequencing

Genomic DNA from each *S. aureus* strain was extracted using QIAGEN DNeasy blood and tissue extraction kit (Hilden, North Rhine-Westphalia, Germany) as per the manufacturer’s instructions. The Nextera XT DNA library preparation kit (Illumina, San Diego, CA, USA) was used to prepare paired-end libraries. All the libraries were multiplexed on one MiSeq run. FastQC version 0.117 (https://www.bioinformatics.babraham.ac.uk/projects/fastqc, accessed on 9 July 2021) was used to assess the quality of sequenced genomes using raw reads. Trimmomatic v0.38 (http://www.usadellab.org/cms/?page=trimmomatic, accessed on 9 July 2021) was used for trimming the adapters from the reads with the setting of minimum read length of 36 and minimum coverage of 15 [69]. De novo assembly using Spades v3.15.0 was performed using the default setting [70]. Assembled genomes were annotated with Prokka v1.12 using GeneBank^®^ compliance flag [71]. The genome of *S. aureus* NCTC 8325 (reference strain in this study) was re-annotated with Prokka to avoid annotation bias.

Multi-locus sequence type (MLST) was determined using PubMLST (https://pubmlst.org/, accessed on 29 September 2021) [72] to find the sequence of each strain. Pan genomes of the *S. aureus* isolates were analyzed using Roary v3.11.2 [73] which uses the GFF3 files produced by Prokka. The program was run using the default settings, which uses BLASTp for all-against-all comparison with 95% of percentage sequence identity. Core genes were taken as the genes which were common in at least 99% of strains. Core genome phylogeny was constructed using Harvest Suite Parsnp v1.2 [74] with *S. aureus* NCTC 8325 (NC_007795.1) as a reference strain. The output file ‘genes_ presence_absence.csv’ generated by Roary was used to compare the *S. aureus* isolates. Phylogenetic tree was constructed using online webtool itol (https://itol.embl.de/, accessed on 4 April 2022). Acquired antibiotic resistance genes of *S. aureus* isolates were examined by using the online database Resfinder v3.1 (https://cge.cbs.dtu.dk/services/ResFinder/, accessed on 30 October 2021) [75]. To determine the association of specific virulence determinants with specific ocular conditions, 128 virulence factors previously described to be associated with many *S. aureus* infections in the virulence factors database (VFDB) were examined (VFDB; Centre for Genomic Epidemiology, DTU, Denmark, http://www.mgc.ac.cn/VFs/main.htm, (accessed on 9 May 2022) respectively [76]. The assembled *S. aureus* isolates were compared to a custom VFDB consisting of 128 virulence genes associated with adhesion, enzymes, immune evasion, type VII secretion systems, and toxins (enterotoxins, enterotoxin-like genes, exfoliative toxins, and exotoxins). A gene sequence had to cover at least 60% of the length of the gene sequence in the database with a sequence identity of 90% to be considered as being present in the strain. As acquired antimicrobial resistance genes may be carried on integrons, *S. aureus* genomes were analyzed for integrons using integron Finder version 1.5.1 (https://bioweb.pasteur.fr/packages/pack@Integron_Finder@1.5.1, accessed on 5 February 2022). There was no evidence for integrons in these 25 isolates. Isolates with the same sequence types were compared for nucleotide similarities using the MUMmer online web tool (http://jspecies.ribohost.com/jspeciesws/#analyse, accessed on 18 February 2022).

### 4.3. Statistical Analysis 

Differences in virulence factors database (VFDB) results for the presence or absence of virulence genes between the disease groups and differences in ocular and non-ocular isolates were analyzed using a chi-square test in GraphPad prism v8.0.2.263 for windows (San Diego, CA, USA, (www.graphpad.com, accessed on 1 June 2022). For all analyses a *p*-value < 0.05 was considered statistically significant and *p*-value < 0.1 was considered as trending towards significance. 

Nucleotide accession: The nucleotide sequences are available in the Genebank under the Bio project accession number PRJNA859391 (https://www.ncbi.nlm.nih.gov/bioproject/PRJNA859391, accessed on 24 July 2022), (genomes accession number (JANHMY000000000, JANHMZ000000000, JANHNA000000000, JANHNB000000000, JANHNC000000000, JANHND000000000, JANHNE000000000, JANHNF000000000, JANHNG000000000, JANHNH000000000, JANHNI000000000, JANHNJ000000000, JANHNK000000000, JANHNL000000000, JANHNM000000000, JANHNN000000000, JANHNO000000000, JANHNP000000000, JANHNQ000000000, JANHNR000000000, JANHNS000000000, JANHNT000000000, JANHNU000000000, JANHNV000000000, JANHNW000000000).

## 5. Conclusions

With respect to virulence determinants distribution, there were some differences between ocular and non-ocular isolates and ocular infectious and niCIE isolates. The current study could not detect plasmids in any of the isolates, as it relied on draft genomes. Further studies including more strains will focus on improvement of the assembly and probe the WGS for possession of pathogenicity islands such as νSaβ, SaPIn2, SaPIl, and SaPIboy. Overall, these findings have extended our understanding of the genomic diversity of *S. aureus* in infectious and non-infectious ocular conditions. The information can be used to elucidate various mechanisms that would help combat virulent and drug resistant strains.

## Figures and Tables

**Figure 1 antibiotics-11-01011-f001:**
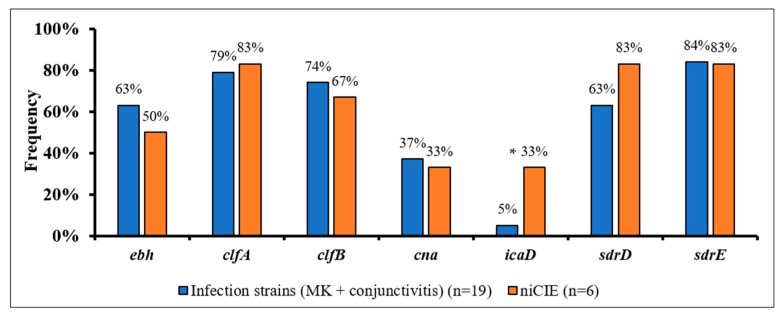
Frequency of seven virulence genes involved in *S. aureus* adhesion by disease group. *, trend to be more common in niCIE strains (*p* = 0.1).

**Figure 2 antibiotics-11-01011-f002:**
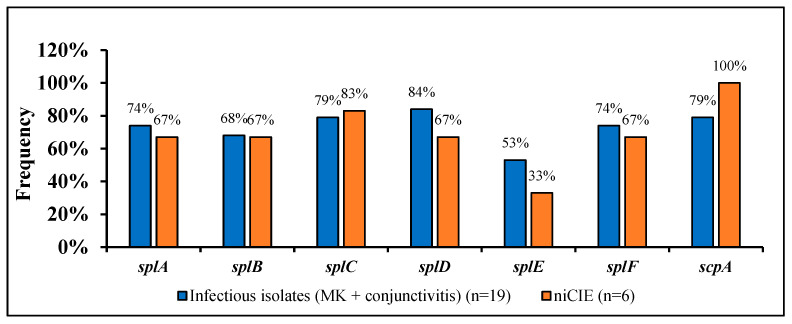
Frequency of 7 proteases in *S. aureus* by disease group.

**Figure 3 antibiotics-11-01011-f003:**
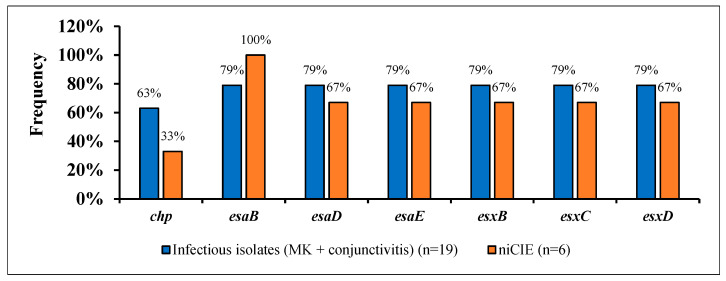
Possession of 7 virulence genes involved in immune evasion and type VII secretion system in *S. aureus* by disease group.

**Figure 4 antibiotics-11-01011-f004:**
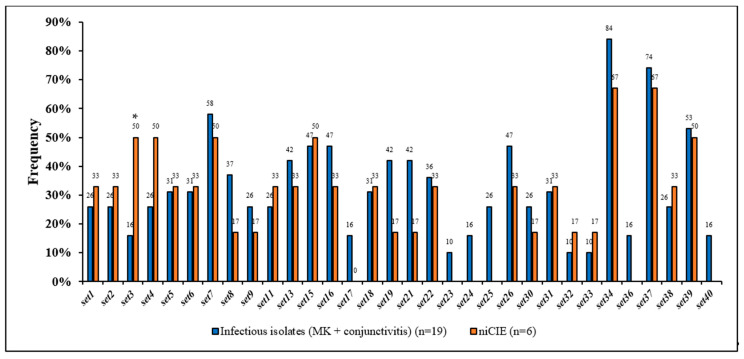
Frequency of 32 enterotoxin-like genes in *S. aureus* by disease group. *, trend more common in niCIE strains (*p* = 0.1).

**Figure 5 antibiotics-11-01011-f005:**
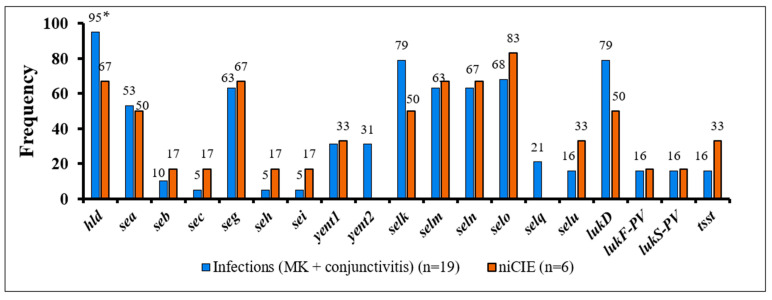
Frequency of 19 enterotoxins, exfoliative toxins and *tsst* in *S. aureus* by disease group. *, trend more common in infectious strains (*p* = 0.1).

**Figure 6 antibiotics-11-01011-f006:**
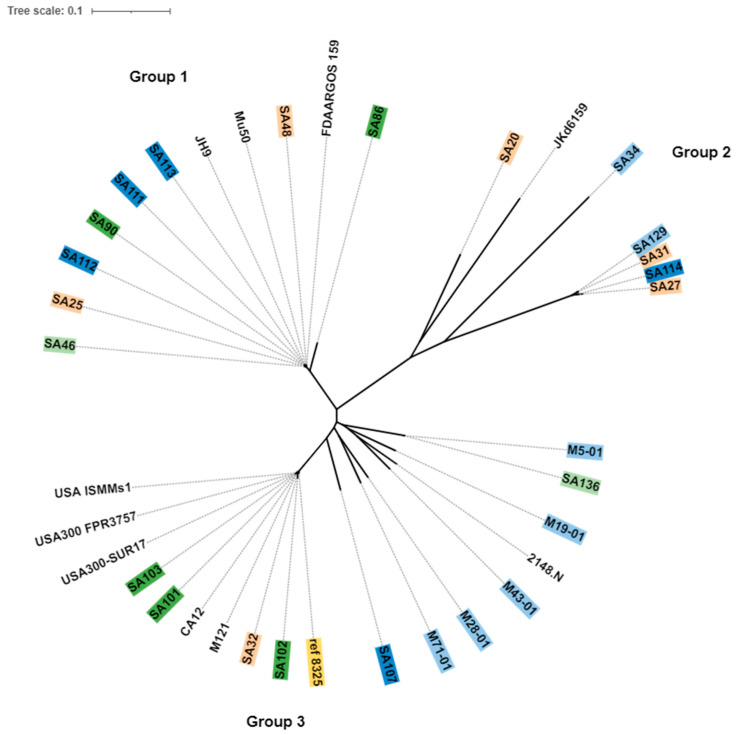
Core genome phylogeny of *S. aureus*, using Parsnp. *S. aureus* strain NCTC 8325, was used as a reference strain (yellow). Isolates highlighted in shades of green indicate conjunctivitis strains; dark green indicates conjunctivitis strains from USA and light green indicates conjunctivitis strains from Australia. Shades of blue indicate MK strains; dark blue represents MK strains from USA and light blue represents MK strains from Australia. The peach color indicates strains from niCIE and strains with no color indicate non-ocular isolates. The tree was constructed using online webtool itol (interactive tree of life, https://itol.embl.de/ (accessed on 4 April 2022).

**Figure 7 antibiotics-11-01011-f007:**
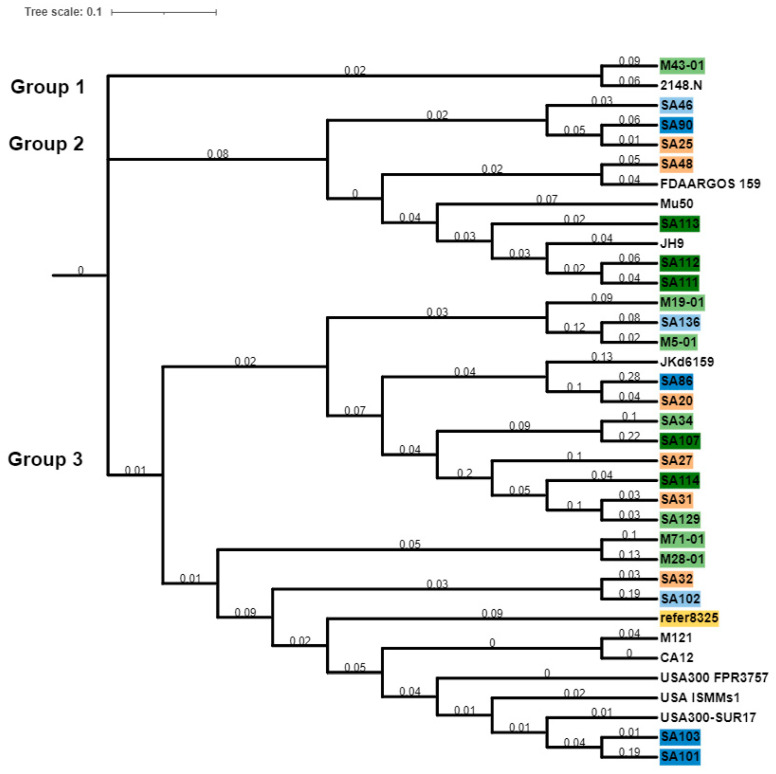
Pan genome phylogeny of *S. aureus* isolates. *S. aureus* strain NCTC 8325 was used as a reference strain (yellow color). Isolates in shades of green are conjunctivitis strains; dark green indicates conjunctivitis strains from USA, and light green conjunctivitis strains from Australia. Shades of blue indicate MK strains; dark blue represents MK strains from USA and light blue represents MK strains from Australia. Isolates in peach color are strains from niCIE and those with no color are non-ocular isolates. The tree was constructed using online webtool itol (interactive tree of life, https://itol.embl.de/, (accessed on 4 April 2022). The tree scale indicates differences between the isolates and branch length indicates the number of changes that have occurred in that branch.

**Table 1 antibiotics-11-01011-t001:** Genetic features of the *S. aureus* isolates.

Ocular Condition	*S. aureus* Isolates	Region	GC Content (%)	No. of Contigs	Total Sequence Length (bp)	CDSs (Total)	tRNAs
**Microbial keratitis**	SA107	USA	32.8	1173	3,599,003	3302	71
	SA111		33	655	3,113,006	2858	85
	SA112		32.9	614	3,170,760	2930	74
	SA113		32.8	530	3,014,859	2771	72
	SA114		32.9	1332	3,175,242	2877	60
	SA34	AUS	32.9	349	2,914,342	2694	60
	SA129		32.9	694	3,105,791	2897	66
	M5-01		32.9	624	2,975,620	2701	85
	M19-01		33	429	2,893,905	2614	77
	M28-01		32.8	475	2,960,866	2715	62
	M43-01		33.1	985	3,029,867	2741	89
	M71-01		32.9	536	2,918,758	2665	74
**Conjunctivitis**	SA86	USA	32.9	3916	4,579,417	3873	76
	SA90		32.8	404	3,015,554	2755	62
	SA101		32.6	998	3,602,977	3296	63
	SA102		32.8	1067	3,406,253	3085	65
	SA103		32.9	479	3,069,147	2857	72
	SA46	AUS	32.9	388	2,903,724	2646	62
	SA136		32.8	735	3,035,909	2803	76
**niCIE**	SA20	AUS	32.8	385	2,909,603	2660	61
	SA25		32.8	366	2,907,754	2622	61
	SA27		32.8	345	2,919,830	2686	67
	SA31		32.8	328	2,976,006	2782	60
	SA32		32.7	649	2,990,036	2665	65
	SA48		32.8	338	2,922,947	2665	64

CDS = coding DNA sequence. Note: all strains had N50 values of 985.

**Table 2 antibiotics-11-01011-t002:** Acquired antimicrobial resistance genes in *S. aureus* isolates from different ocular conditions.

Gene	USA Infectious Isolates (MK+ Conjunctivitis)										Australian Infectious Isolates (MK+ Conjunctivitis)									niCIE					
	107	111	112	113	114	86	90	101	102	103	34	129	M5-01	M19-01	M28-01	M43-01	M71-01	46	136	20	25	27	31	32	48
Beta lactamase resistance gene																									
*blaZ*																									
*mecA*																									
Aminoglycoside resistance gene																									
*aac(6′)*																									
*aph(2* *′)*																									
*ant(6)-la*																									
*aph(3′)-III*																									
*ant(9)-la*																									
*aadD*																									
Macrolide, Lincosamide, Streptogramin B																									
*erm(A)*																									
*msr(A)*																									
*erm(C)*																									
*mph(C)*																									
Tetracycline, chloramphenicol resistance genes																									
*tetK*																									
*cat(pC233)*																									
*tetM*																									
Quaternary ammonium compounds																									
*qacB*																									
*qacD*																									
Pseudomonic acid (Mupirocin)																									
*mupA*																									

Grey color represents the presence of the gene. Dark blue = USA MK strains, light blue = Australian MK strains; dark green = conjunctivitis USA strains, light green = conjunctivitis Australian strain, peach color = niCIE strains.

**Table 3 antibiotics-11-01011-t003:** Sequence types and clonal complexes of *S. aureus* isolates.

*S. aureus* Isolates	Sequence Type	Clonal Complex	Number of:		
			Core Genes	Shell Genes	Pan/Total Genes
107	15	CC15	2392	1187	3579
111	105	CC5	2382	770	3152
112	5	CC5	2380	841	3221
113	105	CC5	2330	782	3112
114	30	CC30	2168	129	3377
86	840	CC5	1984	2577	4561
90	5	CC5	2342	739	3089
101	8	CC8	2533	898	3431
102	8	CC8	2497	760	3257
103	8	CC8	2514	498	3012
34	508	CC45	2194	974	3168
129	34	CC30	2227	1112	3339
M5-01	188	CC1	2267	844	3111
M19-01	NI	NI	2302	684	2986
M28-01	109	CC1	2304	775	3079
M43-01	672	NI	2296	827	3123
M71-01	97	CC97	2315	705	3020
46	5	CC5	2325	664	2989
136	188	CC1	2328	821	3149
20	121	NI	2252	824	3076
25	5	CC5	2341	608	2949
27	39	CC30	2180	996	3176
31	34	CC30	2220	1010	3230
32	8	CC8	2416	501	2917
48	5	CC5	2300	736	3036

NI = not identified. Isolates highlighted in shades of blue indicate MK strains; dark blue represents MK strains from USA and light blue represents MK strains from Australia. Shades of green indicate conjunctivitis strains; dark green indicates conjunctivitis strains from USA and light green indicates conjunctivitis strains from Australia. The peach color indicates strains from niCIE.

**Table 4 antibiotics-11-01011-t004:** Susceptibility and virulence profiles of *S. aureus* strains [12,13].

Ocular Condition	Stain Number	Phenotypic Resistance (R) and Susceptibility (S) Profile	Profile of Virulence Genes Known to Be Possessed by the Isolates
Microbial keratitis USA	SA107	CIP, CEFT, OXA, AZI, POLYB (R) GN, VAN, CHL (S)	*fnbpA*, *eap*, *sspB*, *sspA*, *coa*, *hla*, *hlg*, *hld.*
	SA111	CIP, CEFT, OXA, GN, AZI, POLYB (R) VAN, CHL (S)	*clfA*, *fnbpA*, *eap*, *sspB*, *sspA*, *coa*, *hla*, *hlg*, *hld*
	SA112	CIP, CEFT, OXA, AZI, POLYB (R) GN, VAN, CHL (S)	*clfA*, *fnbpA*, *eap*, *sspB*, *sspA*, *coa*, *hla*, *hlg*, *hld*
	SA113	CIP, CEFT, OXA, AZI, POLYB (R) GN, VAN, CHL (S)	*clfA*, *fnbpA*, *eap*, *sspB*, *sspA*, *coa*, *hla*, *hlg*, *hld*
	SA114	CIP, CEFT, AZI, POLYB (R) GN, VAN, OXA, CHL (S)	*clfA*, *fnbpA*, *eap*, *sspB*, *sspA*, *coa*, *hla*, *hlg*, *hld*
Microbial keratitis Australia	SA34	CEFT, AZI, POLYB (R) CIP, GN, VAN, OXA, CHL (S)	*fnbpA*, *eap*, *scpAsspB*, *sspA*, *coa*, *seb*, *hla*, *hlg*, *hld*
	SA129	CEFT, CHL, AZI, POLYB (R) CIP, GN, VAN, OXA (S)	*clfA*, *fnbpA*, *eap*, *sspB*, *sspA*, *coa*, *hla*, *hlg*, *hld*
	M5-01	CIP, CEFT, CHL, AZI (R) GN, VAN, OXA, POLYB (S)	*clfA*, *fnbpA*, *eap*, *scpA*, *sspB*, *sspA*, *coa*, *hla*, *hlg*, *hld*
	M19-01	CEFT, AZI, POLYB (R) CIP, GN, VAN, OXA, CHL (S)	*fnbpA*, *eap*, *scpA*, *sspB*, *sspA*, *coa*, *hla*, *hlg*, *hld*
	M28-01	CEFT, CHL, AZI, POLYB (R) CIP, GN, VAN, OXA (S)	*clfA*, *fnbpA*, *eap*, *scpA*, *sspB*, *sspA*, *coa*, *hla*, *hlg*, *hld*
	M43-01	CIP, CEFT, OXA, CHL, AZI, POLYB (R) GN, VAN (S)	*clfA*, *eap*, *scpA*, *sspB*, *sspA*, *coa*, *hla*, *hlg*, *hld*
	M71-01	CIP, CEFT, CHL, AZI, POLYB (R) GN, VAN, OXA (S)	*clfA*, *fnbpA*, *eap*, *scpA*, *sspB*, *sspA*, *coa*, *hla*, *hlg*, *hld*
Conjunctivitis USA	SA86	CEFT, CHL, AZI, POLYB (R) CIP, GN, VAN, OXA (S)	*clfA*, *fnbpA*, *eap*, *scpA*, *sspB*, *coa*, *seb*, *hla*, *hlg*, *hld*, *pvl.*
	SA90	CIP, CEFT, AZI, POLYB (R) GN, VAN, OXA, CHL (S)	*clfA*, *fnbpA*, *eap*, *scpA*, *sspB*, *coa*, *hla*, *hlg*, *hld.*
	SA101	CIP, CEFT, OXA, AZI, POLYB (R) GN, VAN, CHL (S)	*clfA*, *fnbpA*, *eap*, *sspB*, *sspA*, *coa*, *hla*, *hlg*, *hld*, *pvl*
	SA102	CIP, CEFT, OXA, AZI, POLYB (R) GN, VAN, CHL (S)	*clfA*, *fnbpA*, *eap*, *sspB*, *sspA*, *coa*, *seb*, *hla*, *hlg*, *hld.*
	SA103	CIP, CEFT, OXA, AZI, POLYB (R) GN, VAN, CHL (S)	*clfA*, *fnbpA*, *eap*, *sspB*, *sspA*, *coa*, *hla*, *hlg*, *hld*, *pvl*
Conjunctivitis Australia	SA46	AZI, POLYB (R) CIP, CEFT, OXA, GN, VAN, CHL (S)	*clfA*, *fnbpA*, *eap*, *scpA*, *sspB*, *sspA*, *coa*, *hla*, *hlg*, *hld.*
	SA136	CIP, CEFT, AZI, POLYB (R) GN, VAN, OXA, CHL (S)	*fnbpA*, *eap*, *scpA*, *sspB*, *sspA*, *coa*, *hla*, *hlg*, *hld.*
niCIE Australia	SA20	CEFT, CHL, AZI, POLYB (R) CIP, GN, VAN, OXA (S)	*fnbpA*, *eap*, *scpA*, *sspB*, *sspA*, *coa*, *seb*, *hla*, *hlg*, *pvl.*
	SA25	AZI, POLYB (R)CIP, CEFT, GN, VAN, OXA, CHL (S)	*clfA*, *fnbpA*, *eap*, *scpA*, *sspB*, *sspA*, *coa*, *seb*, *hla*, *hlg*, *hld.*
	SA27	CEFT, OXA, AZI, POLYB (R) CIP, GN, VAN, CHL (S)	*clfA*, *fnbpA*, *eap*, *scpA*, *sspB*, *sspA*, *coa*, *hla*, *hlg*, *hld.*
	SA31	CIP, CEFT, AZI, POLYB (R) GN, VAN, OXA, CHL (S)	*clfA*, *fnbpA*, *eap*, *scpA*, *sspB*, *sspA*, *coa*, *hla*, *hlg*, *hld.*
	SA32	POLYB (R) CIP, CEFT, AZI, GN, VAN, OXA, CHL (S)	*clfA*, *fnbpA*, *eap*, *scpA*, *sspB*, *sspA*, *coa*, *hla*, *hlg*, *hld.*
	SA48	CEFT, CHL, AZI, POLYB (R) CIP, GN, VAN, OXA (S).	*clfA*, *fnbpA*, *eap*, *scpA*, *sspB*, *sspA*, *coa*, *hla*, *hlg.*

R = resistant, S = sensitive; CIP = ciprofloxacin, CEFT = ceftazidime, OXA = oxacillin, GN = gentamicin, VAN = vancomycin, CHL = chloramphenicol, AZI = azithromycin, POLYB = polymyxin B. *clfA* = clumping factor, *fnbpA* = fibronectin binding protein, *eap* = extracellular adhesion protein, *scpA* = cysteine protease staphopain A, *sspB* = cysteine protease staphopain B, *sspA* = serine protease *v8*, *coa* = collagen binding adhesion, *seb* = enterotoxin, *hla* = alpha-toxin, *hlg* = gamma-toxin, *hld* = delta-toxin, *pvl* = Panton–Valentine leukocidin.

## Data Availability

Data is contained within the article and available upon request.

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
