# Peer review of "Genomics of Staphylococcus aureus Strains Isolated from Infectious and Non-Infectious Ocular Conditions"

_antibiotics, 2022, doi:10.3390/antibiotics11081011_

Round 1
Reviewer 1 Report
Dear Authors
Good Work.
I have some concerns.
Does this paper not need any ethical approval?
The majority of the references utilized are more than 5 years older - this major concern of the paper.
There are a few hundred papers regarding the same issues. My concern is this paper what new adds to science. I hope the authors would clarify this issue. Furthermore, what will be the future utility of this paper?
Regards
Author Response
Dear Editor,
The co-authors and I very much appreciated the encouraging, critical, and constructive comments on this manuscript by the reviewers. We have addressed the comments and made the necessary revisions to the manuscript and wish to re-submit for review.
The reviewer comments and our responses are as follows
Reviewer #1 (Comments for the Author (Required):
Comment 1. Good Work. I have some concerns. Does this paper not need any ethical approval? The majority of the references utilized are more than 5 years older - this major concern of the paper. There are a few hundred papers regarding the same issues. My concern is this paper what new adds to science. I hope the authors would clarify this issue. Furthermore, what will be the future utility of this paper? Regards
Response
We had ethics approval for getting recent isolates from Prince of Wales hospital (MK isolates from Australia) by human research ethics application (HREA), application ID 2020/ETH02783. All other isolates were obtained from culture collection of school of optometry and vision science, UNSW without any identifiable patient data.
There are several, but certainly not a few hundred, papers that have examined S. aureus isolates from keratitis for genes associated with this disease. There is a paucity of information on the virulence factors that may be associated with strains from conjunctivitis and non-infectious corneal infiltrative events. We have added this statement into the introduction – lines 47-50.
We have added the following to the conclusion: lines 436 to 439 “Overall, these findings have extended our understanding of the genomic diversity of S. aureus in infections and non-infectious ocular conditions. The information can be used to elucidate various mechanism that would help combat virulent and drug resistant strains.”
Reviewer 2 Report
The manuscript entitled "Genomics of Staphylococcus aureus strains isolated from infections and non-infectious ocular conditions" is good and in my opinion, it should be accepted in its current form.
Author Response
Response: Thank you for your opinion.
Reviewer 3 Report
Afzal et al report on findings of their investigations involving ocular Staphylococcus aureus isolates using whole genome sequence analysis. It is a follow-up to their recent study that reported on antimicrobial resistance of these strains [10]. Despite investigating a few isolates, the study is of a high significance, as it fills important knowledge gaps on the molecular epidemiology of Staphylococcus aureus.
Moreover, the manuscript is very well written, and the style of presentation makes it clear and easy to follow. There are, however, a few revisions that the authors need to make, following which the manuscript may be accepted for publication.
Line 12: Please rewrite “multi locus” as “multi-locus”; kindly effect this change in all other parts of the manuscript.
Line 14: Please rewrite “data base” as “database”.
Line 22: Please italicize the symbol for probability and introduce spaces between the symbol and the inequality symbol. That is, “p<0.05” could be rewritten as “p < 0.05”; kindly effect this change for all other similar occurrences.
Line 36: Please delete the “the” within “70% of the ocular infections”.
Line 38: Please introduce a hyphen between “life” and “threatening”.
Lines 42–43: Please rewrite “the virulence of strains” as “the virulence of its strains”, “with disease” as “with its disease”, and “used in current” as “used in the current”.
Lines 45: Please rewrite “non-infectious” as “the non-infectious”, “than conjunctivitis” as “than were the conjunctivitis”.
Line 46: Please rewrite “and conjunctivitis” as “and the conjunctivitis”, and “susceptible than MK” as “susceptible to antibiotics than were the MK”. Also, the “MK that begins the next sentence should be preceded with “the”.
Line 59: Please rewrite “genome wide” as “genome-wide”.
Line 89: Please rewrite the caption of Table 2 (Genetic features S. aureus isolates) as “Genetic features of the S. aureus isolates”.
Lines 92–93: Please move the sentence that begins the paragraph to the Methods section, and introduce a hyphen between “Horizontally” and “acquired”.
Lines 123–127: The sentence that begins the paragraph belongs in the Introduction section, and the remainder of that paragraph belongs in the Methods section.
Line 154: Please replace the colon that follows “Similarly” with a comma.
Lines 123–127: This part belongs belongs in the Methods section.
Line 247: Please rewrite “group 2” as “Group 2”.
Line 414–426: The conclusion can be further improved to make the findings more conspicuous. It also needs to be made more concise. For example, the first sentence can be done away with
Author Response
Reviewer #3 (Comments for the Author (Required):
Comment # 1
Line 12: Please rewrite “multi locus” as “multi-locus”; kindly effect this change in all other parts of the manuscript.
Response Rewritten as “multi-locus” in the manuscript.
Comment # 2
Line 14: Please rewrite “data base” as “database”.
Response Changed to database throughout the manuscript.
Comment # 3
Line 22: Please italicize the symbol for probability and introduce spaces between the symbol and the inequality symbol. That is, “p<0.05” could be rewritten as “p < 0.05”; kindly effect this change for all other similar occurrences.
Response Changed throughout the manuscript.
Comment # 4
Line 36: Please delete the “the” within “70% of the ocular infections”.
Response Deleted: “Staphylococcus aureus is responsible for nearly 70% of ocular infections worldwide”
Comment # 5
Line 38: Please introduce a hyphen between “life” and “threatening”.
Response Changed: “S. aureus infections involving the cornea (microbial keratitis; MK) can be sight-threatening and the organism.”
Comment # 6
Lines 42–43: Please rewrite “the virulence of strains” as “the virulence of its strains”, “with disease” as “with its disease”, and “used in current” as “used in the current”.
Response Changed throughout as requested.
Comment # 8
Lines 45: Please rewrite “non-infectious” as “the non-infectious”, “than conjunctivitis” as “than were the conjunctivitis”.
Response Changed as requested.
Comment # 9
Line 46: Please rewrite “and conjunctivitis” as “and the conjunctivitis”, and “susceptible than MK” as “susceptible to antibiotics than were the MK”. Also, the “MK that begins the next sentence should be preceded with “the”.
Response Changed as requested.
Comment # 10
Line 59: Please rewrite “genome wide” as “genome-wide”.
Response Changed as requested
Comment #11
Line 89: Please rewrite the caption of Table 2 (Genetic features S. aureus isolates) as “Genetic features of the S. aureus isolates”.
Response Changed as requested “Table 2. Genetic features of the S. aureus isolates.”
Comment # 12
Lines 92–93: Please move the sentence that begins the paragraph to the Methods section, and introduce a hyphen between “Horizontally” and “acquired”.
Response The sentence that begins the paragraph has been moved to methods section. Please see line 476-478
Acquired antibiotic resistance genes of S. aureus isolates were examined by using only database Resfinder v3.1
Comment # 13
Lines 123–127: The sentence that begins the paragraph belongs in the Introduction section, and the remainder of that paragraph belongs in the Methods section.
Response First sentence of paragraph is moved to introduction section. Please see lines 46-47. S. aureus is known to encode a diverse arsenal of virulence determinants that enables it to cause a variety of infections [10]
Second part of paragraph is moved to methods section. Please see line 482--486.
To determine the association of specific virulence determinants with specific ocular conditions, 128 virulence factors previously described to be associated with many S. aureus infections in the virulence factors database (VFDB) were examined (VFDB; Centre for Genomic Epidemiology, DTU, Denmark, http://www.mgc.ac.cn/VFs/main.htm) respectively.
Comment # 14
Line 154: Please replace the colon that follows “Similarly” with a comma.
Response
Changed as requested
Comment # 15
Lines 123–127: This part belongs belongs in the Methods section.
Response
This part has been moved to methods section. Please see line 482-486.
To determine the association of specific virulence determinants with specific ocular conditions, 128 virulence factors previously described to be associated with many S. aureus infections in the virulence factors database (VFDB) were examined (VFDB; Centre for Genomic Epidemiology, DTU, Denmark, http://www.mgc.ac.cn/VFs/main.htm) respectively.
Comment # 16
Line 247: Please rewrite “group 2” as “Group 2”.
Response Changed as requested
Comment # 17
Line 414–426: The conclusion can be further improved to make the findings more conspicuous. It also needs to be made more concise. For example, the first sentence can be done away with
Response Please see line 436 to 443.
With respect to virulence determinants distribution, there were some differences between ocular and non-ocular isolates and ocular infections and niCIE isolates. The current study could not detect plasmids in any of the isolates, as it relied on draft genomes. Further studies including more strains will focus on improvement of the assembly and probe the WGS for possession of pathogenicity islands such as νSaβ, SaPIn2, SaPIl and SaPIboy. Overall, these findings have extended our understanding of the genomic diversity of S. aureus in infections and non-infectious ocular conditions. The information can be used to elucidate various mechanism that would help combat virulent and drug resistant strains
Reviewer 4 Report
[Antibiotics] Manuscript ID: antibiotics-1826497
This manuscript analysed 25 S. aureus isolates from infections (corneal infection, microbial keratitis, conjunctivitis) and non-infectious corneal infiltrative events (niCIE) isolated from USA and Australia. The whole genome sequencing revealed various antimicrobial resistance genes and virulence factors and the authors compared the prevalence of these genes depending on the conditions of patients. The information on prevalent genes of S. aureus from patients with ocular diseases can give good understandings of the diseases and knowledge of important pathogen in public health field.
No GenBank accession number is available in the manuscript. Please check the ‘Instructions for authors’
- New nucleic acid sequences must be deposited in one of the following databases: GenBank, EMBL, or DDBJ. Sequences should be submitted to only one database.
(suggestion) The term ‘Resistance gene’ is used throughout the manuscript but ‘antimicrobial resistance gene’ should be used for precise explanation.
The style of P value and spacing is not consistent throughout the manuscript
(suggestion) The term ‘Infection strain’ is used in the manuscript. What about changing the word to pathogenic (or infectious) isolates?
(adj.+noun) will be more natural.
L3, L4: Author information is not complete. If “*” is for correspondence, please mention it in detail with contact information. Please check the instructions for authors.
(https://www.mdpi.com/journal/antibiotics/instructions)
L5: UNSW -> The University of New South Wales
L5: NSW -> New South Wales
L15: AUS -> Australia
L18: What type of enzymes are those? Please specify the role of enzymes such as serine protease.
L42: Add reference
L43-45: You used strains from your previous study. The word “current” may mislead the readers. Please change it using “previous“ or something else. (previously reported and used in this study)
L45-47: “More susceptible” is not specific. Please specify the susceptibility of antibiotics you found different with objective numbers (ex: chloramphenicol).
L75: 25 S. aureus isolates from ocular infections and ni-CIEs from USA and Australia.
L89: Table 1
L89 (Table1): All isolates showed no. of contigs over 300. When you used draft genome, you should give N50 value too.
L91: Acquired Resistance Genes -> Antimicrobial Resistance Genes
L92: How do you know they are all “horizontally” transferred? What about chances of mutation or other types of acquired resistance? Delete the ‘horizontally’ or explain it more in detail.
L99: methicillin resistance (MRSA) is a huge public health threat. On what basis could you tell 28% is a small number? Please delete the ‘only’ or use a proper reference to compare to your results.
L101, L102, L105: Please use the same spacing format for p value presentation.
L104-107: Please separate the sentence for better understanding. Comparison of USA and AUS part and the rare genes found in AUS isolates.
L112: Did you get complete sequences of isolates? If it’s not, you should say you could ‘detect’ antimicrobial resistance genes. We cannot tell if the AUS isolates only possessed 5 or if they have more genes.
L112-118: You are only explaining about ‘antimicrobial’ resistance genes. There can be other types of resistance genes such as heavy-metal resistance genes. Please add ‘antimicrobial’.
L115: ‘As acquired resistance genes may be carried on integrons’ -> delete (not for results)
You can explain it in m&m part or discussion part with comprehensive explanation.
L123: This sentence doesn’t belong to result part. Delete the sentence or move it to introduction or discussion part and add a proper reference.
L128-134: Font size is not consistent.
L131-134: In the result part, please explain objective findings. Comprehensive interpretation of findings should be in the discussion part with proper references.
L135: Please delete the ‘small’ for clear and objective explanation. Non-significant differences
L148-L151: Please rephrase the sentences for better understanding.
L167: Please delete ‘small’.
L188 (Figure 4.) Please indicate that numbers are % in Y axis.
Indicate * to each gene if it’s significantly prevalent in one group.
Set24 -> set24
Format should be consistent. Delete one ‘0’ or add all ‘0’ values to others.
L191 (Figure 5.) Indicate * to each gene if it’s significantly prevalent in one group.
L205-206: Can we say 40% and 60% to be ‘commonly’ possessed? Please rephrase the sentences in a way that compares the results.
L212: Please delete the ‘small’.
L217-224:Please check the font size.
L217: 4 -> 4 genes or 4 hemolysin genes
18 -> 18 enterotoxin, enterotoxin-like genes, exfoliative toxins and tsst genes. (or 18 toxin genes)
L219, 221: Please delete the ‘frequently’.
L226-228: Please move the sentence to other parts or delete the sentence.
L230-231: (4%) -> (n=1, 4%)
ST5, 8, 105, 188, 34, 30, 39, 109, 840 were circulating in ocular S. aureus isolates. -> How can you say that? If others found similar STs like yours, explain it in discussion part with proper references. Unless please simply explain the findings.
L233: Is it really new ST? How’s the sequence quality of the strain M19-01? Was it fully sequenced and there was no matching ST in the database or MLST failed since the sequencing of seven housekeeping genes were failed? You can also try traditional method using PCR and sequencing of the genes. If it’s not a new ST, please indicate it as NI.
L236 (Table 4). What is the meaning of colors? Please explain it. (USA, AUS, MK + conjunctivitis, niCIE)
L240: NCTC 8325 -> S. aureus NCTC 8325
L241: Gene bank -> GenBank
L242-251: Can you explain it using Figure 7? Figure 6 may look like redundant.
L244. You need to explain how (method) you made the phylogenetic tree (using Parsnp).
L245: Highly related -> How much were they related? Please use specific number or explain it in objective manner.
L246-247: How did you define XDR and MDR? Reference 10 is abstract and doesn’t show XDR of your strains.
L282: Most -> Most (n=X, X%) …or X% of the isolates…
L253 (Figure 6.): NCTC 8325 -> S. aureus strain NCTC 8325
was used a reference -> was used as a reference strain.
L269 (Figure 7.): NCTC 8325 -> S. aureus strain NCTC 8325
was used a reference -> was used as a reference strain.
How did you make the phylogenetic tree? Please explain the method you used.
L282-284: more susceptible to antibiotics -> to which antibiotics?
If you compare Australian strains with niCIE isolates, it’s hard to say it’s more susceptible. 2 niCIE strains showed tetK but only one Australian infection case showed tetK. Also, Isolate #25 from niCIE group showed many macrolide resistance genes but Australian isolates showed only one resistance gene from one isolate. Please change the sentences.
L289-291: What is the meaning of the different prevalence of hld and hlg? Do they have different contribution in their pathogenesis? Can you find any other report about the prevalence of hld and hlg?
L292: Add reference.
L316-319: Why did you mention the location of aaaD? It’s irrelevant to this study. Please delete the sentence and add interpretation of your finding-aaaD gene in USA isolates.
L320-321: Please reduce the redundant explanation. Delete one (An adhesin / the intercellular adhesion gene).
L326-328: Please add references with numbers (%).
L332: PNAG. Please give the full name of it. It’s first time in the manuscript.
L342-3: Please re-write the sentence for better understanding. (It has a minor role…)
L344: MSCRAMMs. Please give the full name of it. It’s first time in the manuscript.
L349-350: Please re-write the sentence. sdrD may not involved in pathogenesis of currently prevalent types of eye infection, but we don’t know the S. aureus with sdrD could contribute more to eye infection.
L351: conjunctivitis strains -> isolates from conjunctivitis
L352: strains from MK -> isolates from MK
L352: Infection strains from USA -> Isolates from infections from USA
L358-360: may be important during conjunctivitis -> Please re-write the sentence for clear explanation. Ex) May have a role in pathogenesis of conjunctivitis.
L360: the trend for USA infection isolates -> the trend of pathogenic isolates from USA…
L361: Have you searched clonal types of USA isolates? Please add a sentence explaining USA prevalent clonal types with proper references.
L363-366: Could you add a reference for the previous CC or ST data of Australia? Comparison of previous data with your results will be informative.
L369: staphylococci without italic or Staphylococcus spp. with italic.
L371: pathogenicity island SaPIn2 -> Why do you refer SaPIn2 only? pathogeniticy islands such as SaPI1 and SaPIbov can be mentioned using reference 56.
L371-374: An increasing number was reported from one country like Australia or USA? Or the reference paper mentioned that it’s global trend? Please give specific explanation for that.
L376: sea will be Staphylococcal enterotoxin A. Reference #60 found 7 isolates with sea (9.3%) from 75 S. aureus. I couldn’t find the sea prevalence from reference #61. Please change the reference.
L378: strains isolated from -> not italic
L380: patients no ulceration -> patients with no ulceration
L384: multi-drug resistant MDR -> multi-drug resistant (MDR)
L387: MSSA: Please give the full name.
L393: possession of -> possession of genes related to
L397: enzyme genes -> Please specify the function of genes (proteases and …)
L398: toxin genes -> hemolysin genes
L405-407: Please use the specific words, sequence type (STs) and clonal complex (CC). Please use numbers for objective explanation. L406: more common -> were the most prevalent (predominant) types (%, %, respectively…).
L427: Materials and Methods part can be moved to the part between introduction and results. Table 1 will be more natural in that order. Unless, please change the numbers of table.
L428: There should be sub-title for the part. Ex. Bacterial isolates
L433: S. aureus -> Italic
L439: panton valentine leucocidin -> Panton-Valentine leukocidin (PVL)
L449: A de novo -> de novo
L453-454: Please delete the underline. Check the size and type of font.
L453-471: Please add the links for the online-programs such as Resfinder, Virulence Finder, Integron Finder.
L475: chi square test -> chi-square test
Author Response
Reviewer #4 (Comments for the Author (Required):
This manuscript analysed 25 S. aureus isolates from infections (corneal infection, microbial keratitis, conjunctivitis) and non-infectious corneal infiltrative events (niCIE) isolated from USA and Australia. The whole genome sequencing revealed various antimicrobial resistance genes and virulence factors and the authors compared the prevalence of these genes depending on the conditions of patients. The information on prevalent genes of S. aureus from patients with ocular diseases can give good understandings of the diseases and knowledge of important pathogen in public health field.
Comment # 1
No GenBank accession number is available in the manuscript. Please check the ‘Instructions for authors’
- New nucleic acid sequences must be deposited in one of the following databases: GenBank, EMBL, or DDBJ. Sequences should be submitted to only one database.
Response: Please see Line 502 and 503
Supplementary Materials
Nucleotide accession: The nucleotide sequences are available in the Genebank under the Bio project accession number PRJNA859391.
Comment # 2
(suggestion) The term ‘Resistance gene’ is used throughout the manuscript but ‘antimicrobial resistance gene’ should be used for precise explanation.
Response: Changed as requested throughout the manuscript
Comment # 3
The style of P value and spacing is not consistent throughout the manuscript
Response We have now made this consistent.
Comment # 4
(suggestion) The term ‘Infection strain’ is used in the manuscript. What about changing the word to pathogenic (or infectious) isolates?
(adj.+noun) will be more natural.
Response Changed as requested throughout the manuscript as “Infectious isolates”
Comment # 5
L3, L4: Author information is not complete. If “*” is for correspondence, please mention it in detail with contact information. Please check the instructions for authors.
(https://www.mdpi.com/journal/antibiotics/instructions)
Response: Please see lines 5-6
Madeeha Afzal1*, Ajay Kumar Vijay1, Fiona Stapleton1* and Mark D. P. Willcox1*
- School of Optometry and Vision Science, University of New South Wales, Sydney, New South Wales, 2052, Australia
* Authors to whom correspondence should be addressed.
M.A: m.afzal@unsw.edu.au, F.S: f.stapleton@unsw.edu.au, M.D.P.W: m.willcox@unsw.edu.au
Comment # 6
L5: UNSW -> The University of New South Wales
Response: Changed to “University of New South Wales”
Comment # 7
L5: NSW -> New South Wales
Response: Changed as requested
Comment # 8
L15: AUS -> Australia
Response: Changed as requested
Comment # 9
L18: What type of enzymes are those? Please specify the role of enzymes such as serine protease.
Comment # 10
Response: Please see line 20 “serine protease enzymes (splA, splD, splE, splF)”
L42: Add reference
Response:
Reference is added. Please see reference 11 line 541.
Cheung, G.Y.C.; Bae, J.S.; Otto, M.; Pathogenicity and virulence of Staphylococcus aureus. Virulence, 2021 12, 547-569.
Comment # 11
L43-45: You used strains from your previous study. The word “current” may mislead the readers. Please change it using “previous“ or something else. (previously reported and used in this study)
Response: Changed to: “The antibiotic susceptibility data of the isolates previously reported and used in this study”
Comment # 12
L45-47: “More susceptible” is not specific. Please specify the susceptibility of antibiotics you found different with objective numbers (ex: chloramphenicol).
Response: Changed to (lines 52-55): “strains were more susceptible to antibiotics (ciprofloxacin, ceftazidime, oxacillin) than were the conjunctivitis strains, and the conjunctivitis strains were more susceptible to antibiotics (chloramphenicol, azithromycin) than were the MK strains”
Comment # 13
L75: 25 S. aureus isolates from ocular infections and ni-CIEs from USA and Australia.
Response: Changed to: “25 S. aureus isolates from ocular infections and niCIEs from USA and Australia”
Comment # 14
L89: Table 1.
Response: Changed as requested
Comment # 15
L89 (Table1): All isolates showed no. of contigs over 300. When you used draft genome, you should give N50 value too.
Added N50 value as note to the table, please see line 100.
Comment # 16
L91: Acquired Resistance Genes -> Antimicrobial Resistance Genes
Response: Changed as requested
Comment # 17
L92: How do you know they are all “horizontally” transferred? What about chances of mutation or other types of acquired resistance? Delete the ‘horizontally’ or explain it more in detail.
Response: Word horizontally has been removed.
Comment # 18
L99: methicillin resistance (MRSA) is a huge public health threat. On what basis could you tell 28% is a small number? Please delete the ‘only’ or use a proper reference to compare to your results.
Response: Word only has been removed.
Comment # 19
L101, L102, L105: Please use the same spacing format for p value presentation.
Response: Changed as requested
Comment # 20
L104-107: Please separate the sentence for better understanding Comparison of USA and AUS part and the rare genes found in AUS isolates.
Response: Changed to” Resistance gene for tetracycline (tetM) and quaternary ammonium compound (qacD) were found in single isolate (USA) whereas Pseudomonic acid (Mupirocin) was present in only two USA isolates. Whereas quaternary ammonium compound qacB was found in single USA and single Australian isolate. Chloramphenicol resistance gene cat(pC233) was only found in a single Australian isolate only (Table 2).”
Comment # 21
L112: Did you get complete sequences of isolates? If it’s not, you should say you could ‘detect’ antimicrobial resistance genes. We cannot tell if the AUS isolates only possessed 5 or if they have more genes.
Response: Please see lines 121-124.
Overall, in Australian infection isolates only 5 acquired resistance genes were detected, as the current study relied on draft genomes it may not be able to predict actual genomic diversity and could not detect antimicrobial resistance genes, there could be more genes, complete gene sequence of isolates can show actual number of antimicrobial resistance genes.
Comment # 22
L112-118: You are only explaining about ‘antimicrobial’ resistance genes. There can be other types of resistance genes such as heavy-metal resistance genes. Please add ‘antimicrobial’.
Response: Changed as requested throughout the manuscript “antimicrobial resistance”
Comment # 23
L115: ‘As acquired resistance genes may be carried on integrons’ -> delete (not for results)
You can explain it in m&m part or discussion part with comprehensive explanation.
Response: This part of paragraph has been moved to methodology, please see line 492 to 495.
As acquired antimicrobial resistance genes may be carried on integrons, S. aureus genomes were analysed for integrons using integron Finder version 1.5.1 (https://bioweb.pasteur.fr/packages/pack@Integron_Finder@1.5.1). There was no evidence for integrons in these 25 isolates.
Comment # 24
L123: This sentence doesn’t belong to result part. Delete the sentence or move it to introduction or discussion part and add a proper reference.
Response: Sentence has been moved to introduction section, please see line 46.
- aureus is known to encode a diverse arsenal of virulence determinants that enables it to cause a variety of infections. [10].
Comment # 25
L128-134: Font size is not consistent.
Response: We have now made this consistent.
Comments # 26
L131-134: In the result part, please explain objective findings. Comprehensive interpretation
of findings should be in the discussion part with proper references.
Response: Comprehensive interpretation in removed from results section. Please see lines 134-138
Of the 128 virulence factors examined, 22 virulence genes (atl, ebh, clfA, clfB, cna, ebp, eap, efb, fnbA, fnbB, icaA, icaB, icaC, icaD, icaR, sdrC, sdrD, sdrE, sdrF, sdrG, sdrH, spa) in VFDB are categorised as genes involved in S. aureus adhesion. Of these adhesins, atl, ebp, eap, efb, fnbA, fnbB, icaA, icaB, icaC, icaR, sdrC and spa were found in ≥96% of all S. aureus isolates. On the other hand, sdrF, sdrG, sdrH were not detected in any of the strains.
Comment # 27
L135: Please delete the ‘small’ for clear and objective explanation. Non-significant differences
Response: Changed as requested “showed non-significant differences”
Comment # 28
L148-L151: Please rephrase the sentences for better understanding.
Response: Sentence is rephrased, please see line 155-158 “The isolates from infections or niCIEs did not show significant differences (p > 0.05) or trend towards significance (p > 0.1), for the possession of other seven proteases, (Figure 2)”.
Comment # 29
L167: Please delete ‘small’.
Response: Changed as requested. Figure 3 shows the differences in possession
Comment # 30
L188 (Figure 4.) Please indicate that numbers are % in Y axis.
Indicate * to each gene if it’s significantly prevalent in one group.
Set24 -> set24
Format should be consistent. Delete one ‘0’ or add all ‘0’ values to others.
Response: % in Y axis is added please see Figure 4 and 5. We have now made this consistent, (p = 0.1). Please see figure 4 and 5 shows infections and niCIE group, * is added with set 3 in figure4, and hld in figure 5 for significant differences in infectious and niCIE isolates.
Figure 4
Please see lines 182-187
Of these the only significant differences or trends for differences were: niCIE isolates tended to have a higher frequency (50%) of only set3 (p = 0.1) (Figure 4) than infection strains (16%), and infection strains tended to have a higher frequency (95%) of only hld (p = 0.1) than niCIE (67%) (Figure 5).
Comment # 31
L191 (Figure 5.) Indicate * to each gene if it’s significantly prevalent in one group.
|
Please see lines 191-195
Of these the only significant differences or trends for differences were: niCIE isolates tended to have a higher frequency (50%) of only set3 (p = 0.1) (Figure 4) than infection strains (16%), and infection strains tended to have a higher frequency (95%) of only hld (p = 0.1) than niCIE (67%) (Figure 5).
Comment # 32
L205-206: Can we say 40% and 60% to be ‘commonly’ possessed? Please rephrase the sentences in a way that compares the results.
Response: Please see line 214-217
- aureus ocular isolates showed higher frequency for the possession of cna (40% vs 0%; p = 0.03) and icaR (100% vs 70%; p = 0.01) whereas non-ocular isolates showed higher frequency for the possession of icaD (60% vs 12%; p = 0.007), ebh (90% vs 60%; p = 0.1) and sdrD (100% vs 68%; p = 0.07).
Comment # 33
L212: Please delete the ‘small’.
Response: Changed as requested “with non-significant differences”
Comment # 34
L217-224: Please check the font size.
Response:
We have now made this consistent.
Comment # 35
L217: 4 -> 4 genes or 4 hemolysin genes
18 -> 18 enterotoxin, enterotoxin-like genes, exfoliative toxins and tsst genes. (or 18 toxin genes)
Response: Changed as requested, “4 hemolysin genes (hla, hlgA, hlgB, hlgC)”.
Line 219, 18 toxin genes
Comment # 36
L219, 221: Please delete the ‘frequently’.
Response: Changed as requested “Of the remaining 52 toxins, S. aureus ocular isolates possessed sea (80% vs 20%; p = 0.001), set1 (28% vs 0%; p = 0.08), set5 (32% vs 0%; p = 0.07), set16 (44% vs 0%; p = 0.01), set19 (36% vs 0%; p = 0.03), whereas non-ocular isolates possessed”
Comment # 37
L226-228: Please move the sentence to other parts or delete the sentence.
Response: The sentence is removed
The multi-locus sequence typing (MLST) of seven housekeeping genes allowed classification into sequence types (STs) and closely related STs were grouped into clonal complexes (CC) [26]; for example, ST5 ST105 and ST840 were grouped in CC5.
Comment # 38
L230-231: (4%) -> (n=1, 4%)
ST5, 8, 105, 188, 34, 30, 39, 109, 840 were circulating in ocular S. aureus isolates. -> How can you say that? If others found similar STs like yours, explain it in discussion part with proper references. Unless please simply explain the findings.
Response: Findings are simply explained, please see line 237-238. “ST5 (n=5, 20%) and ST8 (n=4, 16%) were the most common sequence types in this cohort of ocular isolates”
Comment # 39
L233: Is it really new ST? How’s the sequence quality of the strain M19-01? Was it fully sequenced and there was no matching ST in the database or MLST failed since the sequencing of seven housekeeping genes were failed? You can also try traditional method using PCR and sequencing of the genes. If it’s not a new ST, please indicate it as NI.
Response: Changed as requested, please see line 239 “For strain M19-01, no ST type was identified, and it was named as NI (Table 4)”
Comment # 40
L236 (Table 4). What is the meaning of colors? Please explain it. (USA, AUS, MK + conjunctivitis, niCIE)
Response: Please see lines 243-237 and lines 263-267.“Isolates highlighted in shades of blue indicate MK strains; dark blue represents MK strains from USA and light blue represents MK strains from Australia. Shades of green indicates conjunctivitis strains; dark green indicates conjunctivitis strains from USA and light green indicates conjunctivitis strains from Australia. The peach colour indicates strains from niCIE.”
Comment # 41
L240: NCTC 8325 -> S. aureus NCTC 8325
Response: Changed as requested “S. aureus NCTC 8325 (NC_007795.1)”
Comment # 42
L241: Gene bank -> GenBank
Response: Changed as requested “Genebank”
Comment # 43
L242-251: Can you explain it using Figure 7? Figure 6 may look like redundant.
Response: Thanks for your comment, but figure 6 explain core genes of isolates and figure 7 explains pan genes.
Comment # 44
L244. You need to explain how (method) you made the phylogenetic tree (using Parsnp).
Response: Please see line 479,480 in the methods section. “Phylogenetic tree was constructed using online webtool itol (https://itol.embl.de/)”
Comment # 45
L245: Highly related -> How much were they related? Please use specific number or explain it in objective manner.
Response: Please see line 245-247.“Isolates in Group 1 were related, as they belonged to same CC5. This group also contained all the extensively-drug resistant isolates (XDR) (SA111, SA112, SA113) and three multi-drug resistant (MDR) isolates (SA90, SA48, SA46) reported in the previous study”
Comment # 46
L246-247: How did you define XDR and MDR? Reference 10 is abstract and doesn’t show XDR of your strains.
Response: XDR and MDR are defined in previous study correct reference of paper is added now. Please see reference 12 line 544. Please see lines 256-258. “(XDR: resistant to almost all antibiotics) (SA111, SA112, SA113) and three multi-drug resistant (MDR: resistant to three different classes of antibiotics) isolates (SA90, SA48, SA46)”
Comment # 47
L282: Most -> Most (n=X, X%) …or X% of the isolates…
L253 (Figure 6.): NCTC 8325 -> S. aureus strain NCTC 8325
was used a reference -> was used as a reference strain.
Response: Changed as requested, please see line 295 “Most (n=22, 88%) of the isolates used in the study were MDR”
Changed as requested“S. aureus strain NCTC 8325 was used as a reference strain”.
Comment # 48
L269 (Figure 7.): NCTC 8325 -> S. aureus strain NCTC 8325
was used a reference -> was used as a reference strain.
How did you make the phylogenetic tree? Please explain the method you used.
Response: Changed as requested “S. aureus strain NCTC 8325 was used as a reference strain”.
Method for phylogenetic tree constructed has been added please see line 286-287.
“The tree was constructed using online webtool itol (interactive tree of life, https://itol.embl.de/)”.
Comment # 49
L282-284: more susceptible to antibiotics -> to which antibiotics?
If you compare Australian strains with niCIE isolates, it’s hard to say it’s more susceptible. 2 niCIE strains showed tetK but only one Australian infection case showed tetK. Also, Isolate #25 from niCIE group showed many macrolide resistance genes but Australian isolates showed only one resistance gene from one isolate. Please change the sentences.
Response: Please see line 295-297, which reports phenotypic susceptibility data of the previous study.
Phenotypically non-infectious niCIE isolates in previous study were more susceptible to antibiotics (ciprofloxacin, oxacillin) than conjunctivitis and MK strains.
Comment # 50
L289-291: What is the meaning of the different prevalence of hld and hlg? Do they have different contribution in their pathogenesis? Can you find any other report about the prevalence of hld and hlg?
Response: The difference prevalence is the previous study reference [13], showed prevalence in hld and hlg, with current study because previous study used PCR for 63 S. aureus strains from infections and niCIE conditions, whereas current study used only 25 S. aureus strain for WGS analysis, the difference in prevalence is because of a smaller number of isolates used in the current study.
Comment # 51
L292: Add reference.
Response: Please see line 296, reference 12
Afzal, M.; Vijay, A.K.; Stapleton, F.; Willcox, M. Virulence genes of Staphylococcus aureus associated with keratitis, conjunctivitis and contact lens-assocaied inflammation. Transl Vis Sci Technol 2022, 11, 5
Comment # 52
L316-319: Why did you mention the location of aaaD? It’s irrelevant to this study. Please delete the sentence and add interpretation of your finding-aaaD gene in USA isolates.
Response: Location of aaaD gene has been removed
Comment # 53
L320-321: Please reduce the redundant explanation. Delete one (An adhesin / the intercellular adhesion gene).
Response: Changed as requested, please see line 322-323
“Strains from niCIE showed a trend of higher frequency of possession icaD, the intercellular adhesion gene”
Comment # 54
L326-328: Please add references with numbers (%).
Response: Changed as requested, please see lines 336-330.
In the current study when ocular isolates were compared to non-ocular isolates, showed higher frequency for the possession of cna (40% vs 0%) and icaR (100% vs 70%), whereas non-ocular isolates showed higher frequency for the possession of icaD (60% vs 12%), ebh (90% vs 60%) and sdrD (100% vs 68%).
Comment # 55
L332: PNAG. Please give the full name of it. It’s first time in the manuscript.
Response: Changed as requested, please see line 344
PNAG (Poly-N-acetylglucosamine) production.
Comment # 56
L342-3: Please re-write the sentence for better understanding. (It has a minor role…)
Response: Please see line 354-355. “The lower frequency of ebh possession in S. aureus ocular strains suggests it has a minor role in eye infections”.
Comment # 57
L344: MSCRAMMs. Please give the full name of it. It’s first time in the manuscript.
Response: Changes as requested, please see line 356. “MSCRAMMs (Microbial surface components recognizing adhesive matrix molecules)”
Comment # 58
L349-350: Please re-write the sentence. sdrD may not involved in pathogenesis of currently prevalent types of eye infection, but we don’t know the S. aureus with sdrD could contribute more to eye infection.
Response: Sentence is re-written as suggested, please see line 364-366. “The lower frequency of possession of this gene in ocular isolates indicates sdrD may not be involved in pathogenesis of currently prevalent types of eye infections, however, S. aureus with sdrD could contribute more to eye infection”
Comment # 59
L351: conjunctivitis strains -> isolates from conjunctivitis
Response: Changed as requested, please see line 367 “Isolates from conjunctivitis”
Comment # 60
L352: strains from MK -> isolates from MK
Response: Changed as requested, please see line 368 “Isolates from MK”
Comment # 61
L352: Infection strains from USA -> Isolates from infections from USA
Response: Changed as requested, please see line 368 “Isolates from infections from USA”
Comment # 62
L358-360: may be important during conjunctivitis -> Please re-write the sentence for clear explanation. Ex) May have a role in pathogenesis of conjunctivitis.
Response: Changed as requested, please see line 375 “may have a role in pathogenesis of conjunctivitis”
Comment # 63
L360: the trend for USA infection isolates -> the trend of pathogenic isolates from USA…
Response: Changed as requested, please see line 376 “the trend of pathogenic isolates from USA infections to possess”
Comment # 64
L361: Have you searched clonal types of USA isolates? Please add a sentence explaining USA prevalent clonal types with proper references.
Response: Reference showing prevalent clonal types in USA ocular S. aureus has been added, please see line 378 and 379.
“Studies reported ST5 (27%), ST8 (16%), ST30 (9%) and ST45 (6%) as prevalent clonal types in USA ocular isolates [54]”.
Reference # 54, line 642
Johnson, W.L.; Sohn, M.B.; Taffner, S.; Chatterjee, P.; et al. Genomics of Staphylococcus aureus ocular isolates. PloS one, 2021 16, e0250975.
Comment # 65
L363-366: Could you add a reference for the previous CC or ST data of Australia? Comparison of previous data with your results will be informative
Response: Could not find reference to CC or ST of ocular S. aureus of Australia, but a reference of prevalent CC in Australia is added, please see line 379 and 380 reference #55.
Ng, J.W.; Holt, D.C.; Lilliebridge, R.A.; Stephens, A.J.; Huygens, F., et al. Phylogenetically distinct Staphylococcus aureus lineage prevalent among indigenous communities in northern Australia. J Clin Microbio, 2009. 47, 2295-2300.
Comment # 66
L369: staphylococci without italic or Staphylococcus spp. with italic.
Response: Changed as requested, please see line 386 “staphylococci”
Comment # 67
L371: pathogenicity island SaPIn2 -> Why do you refer SaPIn2 only? pathogeniticy islands such as SaPI1 and SaPIbov can be mentioned using reference 56.
Response: Changed as suggested, please see line 388 “SaPIl and SaPIboy”
Comment # 68
L371-374: An increasing number was reported from one country like Australia or USA? Or the reference paper mentioned that it’s global trend? Please give specific explanation for that.
Response: The increasing number was not reported from one country like Australia or USA, it’s a global trend that Around 80% of S. aureus isolates, including commensal, clinical, and food-poisoning isolates, carry an average of 5–6 SE genes.
Please see line 391-393 “An increasing number of enterotoxins and enterotoxin-like genes in S. aureus have been identified and it’s a global trend that around 80% of S. aureus both pathogenic and non-pathogenic isolates carry average of 5-6 enterotoxin genes [56-58]”
Comment # 69
L376: sea will be Staphylococcal enterotoxin A. Reference #60 found 7 isolates with sea (9.3%) from 75 S. aureus. I couldn’t find the sea prevalence from reference #61. Please change the reference.
Response: Please see line 398.
“However, the current study findings are not consistent with earlier studies [59, 60]”
Comment # 70
L378: strains isolated from -> not italic
Response: Changed as requested “S. aureus strains isolates”
Comment # 71
L380: patients no ulceration -> patients with no ulceration
Response: Changed as requested, please see line 398 “patients with no ulceration”
Comment # 72
L384: multi-drug resistant MDR -> multi-drug resistant (MDR)
Response: Changed as requested, please see line 403 “were multi-drug resistant (MDR)”
Comment # 73
L387: MSSA: Please give the full name.
Response: Changed as requested, please see line 403 “MSSA (methicillin sensitive S. aureus)”
Comment # 74
L393: possession of -> possession of genes related to
Response: Changed as requested “possession of genes related to”
Comment # 75
L397: enzyme genes -> Please specify the function of genes (proteases and …)
Response: Changed as requested, please see line 416-417 “the enzyme genes (proteases, thermonuclease, lipase, staphylokinase and hyaluronate lyase)”
Comment # 2
L398: toxin genes -> hemolysin genes
Response: Changed as requested “ hemolysin genes”
Comment # 76
L405-407: Please use the specific words, sequence type (STs) and clonal complex (CC). Please use numbers for objective explanation. L406: more common -> were the most prevalent (predominant) types (%, %, respectively…).
Response: Changed as requested, please see line 429-431
sequence types (STs), ST5 (20%), ST8 (16%), clonal complex (CC), CC30 (16%) and CC1 (12%) were the most prevalent (predominant) types respectively.
Comment # 77
L427: Materials and Methods part can be moved to the part between introduction and results. Table 1 will be more natural in that order. Unless, please change the numbers of table.
Response: Tables numbers has been changed as requested.
Comment # 78
L428: There should be sub-title for the part. Ex. Bacterial isolates
Response: Subtitle has been added please see lines 446 “Bacterial isolates”
Comment # 79
L433: S. aureus -> Italic
Response: Changed as requested
Comment # 80
L439: panton valentine leucocidin -> Panton-Valentine leukocidin (PVL)
Response: Changed as requested, please see lines 458 “Panton-Valentine leukocidin"
Comment # 81
L449: A de novo -> de novo
Response: Changed as requested, please see lines 458 “de novo”
Comment # 82
L453-454: Please delete the underline. Check the size and type of font.
Response: Changed as requested, please see lines 472-473. We have made it consistent.
“PubMLST (https://pubmlst.org/) [23] to find the sequence of each strain”.
Comment # 83
L453-471: Please add the links for the online-programs such as Resfinder, Virulence Finder, Integron Finder.
Response: Links for the online programs such as Resfinder, Virulence Finder, Integron Finder have been added, please see lines 483, 487, 495
“Resfinder v3.1 (https://cge.cbs.dtu.dk/services/ResFinder/).”
“VFDB; Centre for Genomic Epidemiology, DTU, Denmark, http://www.mgc.ac.cn/VFs/main.htm).”
“Finder version 1.5.1 (https://bioweb.pasteur.fr/packages/pack@Integron_Finder@1.5.1).”
Comment # 84
L475: chi square test -> chi-square test
Response: Changed as requested, please see line 502 “chi-square test”